# Trapped in my inner prison—Cross-sectional examination of internal and external entrapment, hopelessness and suicidal ideation

**Inken Höller\*, Amelie Kremers, Dajana Schreiber, Thomas Forkmann**

Department of Clinical Psychology, University of Duisburg-Essen, Essen, Germany

\* inken.hoeller@uni-due.de

## Abstract

### Background

Within the integrated motivational-volitional model of suicidal behavior, entrapment that consecutively leads to hopelessness is considered as a proximal risk factor for suicidal ideation. Entrapment can refer to both external and internal circumstances whereby results of recent studies indicate that internal entrapment plays a more important role than external entrapment in the development of suicidal ideation. It has been considered that to escape internal entrapment might be more complicated than to change external circumstances. However, it remains unclear whether the greater effect of internal entrapment on suicidal ideation is due to greater feelings of hopelessness. Therefore, the aim of this study was to address this research gap and to examine the effects of internal and external entrapment on hopelessness and suicidal ideation.

### Methods

N = 454 participants from a community sample (75% female) aged between 18 and 73 years ($M = 29.91$, $SD = 11.56$) conducted a cross-sectional online survey. All participants were assessed for suicidal ideation, hopelessness, and internal as well as external entrapment. Pearson product-moment correlations and two mediation analyses were conducted.

### Results

All constructs were significantly correlated. For both internal and external entrapment, an effect on suicidal ideation was found. Both effects were partially mediated by hopelessness, this mediation was larger for external entrapment. The completely standardized indirect effect used to compare the mediation models was larger for external entrapment than for internal entrapment.

**Data Availability Statement:** All relevant data are within the Supporting Information files.

**Funding:** The authors received no specific funding for this work.

**Competing interests:** The authors have declared that no competing interests exist.

**Abbreviations:** IMV model, Integrative Motivational-Volitional Model of Suicidal Behavior; IE, Internal Entrapment; EE, External Entrapment.

## Conclusions

Hopelessness mediated the association between external entrapment as well as internal entrapment and suicidal ideation. This effect was larger for external entrapment.

## Background

Everyone knows the feeling of being trapped due to external or internal circumstances [1]. The loss of one's job, an argument in a relationship, or financial problems can lead to the feeling that there is no way out and that we are caught in a situation. The same applies to internal circumstances, for example, through thoughts that we believe we cannot control. Entrapment has been associated with depression [2] but has also come to the fore as a potentially important proximal risk factor for suicidal ideation. One of the current ideation-to-action theories that seeks to explain the development of suicidal ideation and the transition to suicidal behavior is the Integrative Motivational-Volitional Model [3]. The model consists of three phases whereby the first phase focuses on the individual's biopsychosocial context, a person's genetic, cognitive vulnerabilities, and a person's negative life events as well as environmental influences. The second phase proposes that feelings of defeat and humiliation lead to entrapment, which in turn facilitates suicidal ideation. Additional "threat-to-self" as well as "motivational" moderators influence the transitions from defeat to entrapment and from entrapment to suicidal ideation. The third phase of the IMV model describes the actual transition from suicidal ideation to suicidal behavior. This transition is again influenced by "volitional" moderators such as the capability for suicide [4].

Within the frame of the IMV model, entrapment plays a central role in the development of suicidal ideation. This central role of entrapment could be shown by a tremendous amount of research. Branley-Bell, O'Connor [5] showed in a prospective study with N = 299 adult participants that defeat had an indirect effect on suicidal ideation through entrapment. Park, Cho [6] pointed to entrapment as a risk factor–even superior to relatively robust risk factors such as depression–for suicidal ideation in a cross-sectional study with more than 11,000 students from South Korea. The effect of entrapment on suicidal ideation and the subordinated role of defeat could also be demonstrated by several other studies [7–10].

Even though Gilbert and Allan [1] originally proposed entrapment as a two-dimensional construct, the IMV model does not explicitly differentiate between internal and external entrapment. However, Forkmann, Teismann [11] demonstrated by means of network analysis the two dimensional structure of entrapment. Recent research additionally revealed that internal entrapment seems to be more important in the development of suicidal ideation than external entrapment [9]. Owen, Dempsey [12] verified in their prospective longitudinal designed study, that the subtypes of entrapment have differential effects on suicidal ideation. Only internal but not external entrapment mediated the relationship between defeat and suicidal ideation in a sample of 80 adults with bipolar disorder. Höller, Rath [13] showed in a prospective study with a sample at high-risk of suicide that only internal entrapment could predict suicidal ideation but not external entrapment. O'Connor and Portzky [9] and Rasmussen, Fraser [8] stated that internal entrapment might be more difficult to modify deliberately for an individual than external entrapment, which then could lead in turn to more intense feelings of hopelessness which then entail heightened suicidal ideation. Contrary, Lucht, Höller [14] demonstrated a simple mediation of the relationship between defeat and suicidal ideation via both internal and external entrapment. So far, it remains unclear how, why and to what extent internal and external entrapment influence suicidal ideation.

A construct that appears to be closely related to entrapment is hopelessness, which also has been shown to be a risk factor for suicidal ideation [15]. The Cry of Pain Modell [16] assumes that insufficient escape routes or options for action lead to feelings of entrapment, which then might lead to hopelessness because there seems to be no way out of the circumstances. As a consequence this can then lead to suicidal ideation. Johnson, Gooding [17] described that hopelessness develops from entrapment. Since O'Connor and Kirtley [3] only exemplarily name motivational moderators and additionally emphasize that moderators might also function as mediators of the proposed relations between the concepts in their model, hopelessness could be a variable that mediates the relationship between entrapment and suicidal ideation in the motivational phase of the model. This assumption is substantiated by findings of Littlewood, Gooding [18] showing that hopelessness mediates the association between entrapment and suicidality, but they did not differentiate between internal and external entrapment. Individuals might experience internal and/or external entrapment, which leads to hopelessness and then to suicidal ideation.

Even though, due to the different effects of internal and external entrapment on suicidal ideation that have been reported empirically in the mentioned studies, a differential impact of these constructs on hopelessness seems plausible considering the idea that the escape from external circumstances might be easier [8, 9]. However, this has not yet been investigated. Thus, we hypothesized that 1) internal and external entrapment, hopelessness and suicidal ideation correlate significantly but that the correlations are greater for internal than for external entrapment and 2a) both the effect of internal and external entrapment on suicidal ideation is partially mediated by hopelessness but 2b) this association is greater mediated by hopelessness for internal entrapment.

## Methods

### Procedure

Data was collected from April 2021 to July 2021 in a cross-sectional online survey that was conducted on www.soscisurvey.com. Participants were recruited through advertisement at the University of Duisburg-Essen as well as social media (e.g., Facebook). Before starting the survey, participants were informed about the purpose of the study, the voluntary nature of participation, data storage and security, and gave informed consent by accepting the conditions on the first page of the online survey. In case participants did not agree, the participation in the study was not possible and the window was closed. Inclusion criteria were an age above 18 years and sufficient knowledge of the German language. In addition, only those persons who answered the attention check item correctly were considered in the analysis. The attention check item asked participants to check the answer option "4". In case they did not check this item correctly due to inattentiveness, they were excluded from analyses.

The study was approved by the Ethic Committee of the University of Duisburg-Essen and was in accordance with the Declaration of Helsinki [19].

### Measures

In order to gain insight into any past and current mental disorders of the participants, they were asked with one item each whether they had a mental disorder in the past or at the moment and whether they were receiving treatment.

In the following, all measures relevant for this study will be reported.

**Internal and external entrapment.** To assess internal and external entrapment, the entrapment scale (ES; [1]; German version: [20]) was used. The ES consists of 16 items, whereby six items assess internal entrapment ("I would like to escape from my thoughts and

feelings") and ten items assess external entrapment ("I am in a relationship I cannot get out of") referring to the last week. Each item must be answered on a 5-point Likert Scale ranging from 0 = not at all to 4 = very strong. A sum score was calculated for each of the two subscales. Internal consistencies have been shown to be good (Cronbach's $\alpha \geq$ .84) [1, 20]. For this sample, internal consistency was excellent for both internal (Cronbach's $\alpha$ = .96) and external entrapment. (Cronbach's $\alpha$ = .91).

**Suicidal ideation.** Suicidal ideation has been measured using the Beck Scale for Suicidal Ideation (BSS; [21]; German version: [22]), a 21-item self-report measure assessing suicidal ideation (SI) on a 3-point scale from 0 to 2 with differing item phrasing. When participants answered items 4 and 5 (screening items) with at least 1, the first 19 items were used to build a sum score. Otherwise, participants had a sum score of zero. The items 20 and 21 describe intensity and frequency of former suicide attempts and are not part of the sum score. Since there is an ongoing discussion on the factor structure of the BSS [23], only the sum of the first five items were used as a screener for suicidal ideation (BSS-Screen), which has been shown to have good scale properties and excellent internal consistency (Cronbach's $\alpha$ = .97) [24]. In this sample, internal consistency was also excellent with Cronbach's $\alpha$ = .91.

**Hopelessness.** The Beck Hopelessness Scale (BHS; [25]; German version: [26]) includes 20 true-false items that assess pessimistic and hopeless cognitions ("I might as well give up, because there is nothing I can do to improve the situation"). The BHS is also evaluated via a sum score, while higher scores indicate higher levels of hopelessness. Validity and reliability of the German version of the BHS has been shown in prior studies [26]. Internal consistency in the current sample was excellent with Cronbach's $\alpha$ = .92.

## Statistical analyses

All statistical analyses were conducted with SPSS version 27.0. For the first hypothesis that all constructs are correlated but that correlations are higher for internal than for external entrapment, person product-moment correlations were calculated. Correlations $\geq$ 30 were considered as moderate and correlations $\geq$ .50 were considered as strong correlations [27]. To test whether the correlation coefficients significantly differed, correlation coefficients were adjusted with Fisher's z transformation and the calculator of Hemmerich [28] was used. Since the correlation coefficients were based on the same sample and had the same variable (BSS-Screen) in common, the significance for dependent groups with a third variable was calculated. The z-scores were compared with the critical z-score of 1.645 (5%-level for one-sided testing). For a significant difference, the z-score had to be higher than the critical z-score.

For the second hypothesis, two mediation analyses were conducted using SPSS-Macro PROCESS version 3.5 [29]. Mediation model 1 included internal entrapment as the predictor (X), hopelessness as the mediator (M) and suicidal ideation as the dependent variable (Y). Mediation model 2 only differed in including external instead of internal entrapment as the predictor (X). The total effect (the sum of direct and indirect effects ($c$), the indirect effect, which is the product of the effect of X on M ($a$) as well as of M on Y ($b$) were examined [30]. The effect from X on Y in consideration of M ($c'$) was examined. We expected this effect to be at least smaller than $c$ (partial mediation). Additionally, we examined whether the indirect effect ($ab$) became significant [31]. We differentiated between the direct effect from X to Y ($c'$), the indirect effect ($ab$) and the total effect ($c = c'+ab$) (see Fig 1). Post- hoc, we conducted a third mediation model with total entrapment (internal and external entrapment as the predictor). Results of this can be found in the Fig 1, Tables 2 and 3 in S1 File.

For all analyses, 5,000 bootstrap iterations were selected. For all effects, completely standardized effect sizes (CS), which express the respective effects in units of standard deviation,

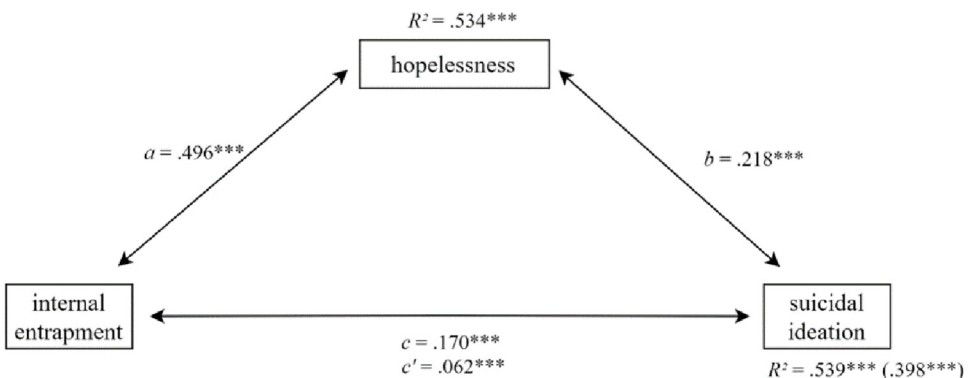

**Fig 1. Mediation model 1 with internal entrapment as predictor.**

were additionally calculated. Higher values correspond to larger effects. The determination coefficient $R^2$ was also calculated. $R^2$ expresses, how much of the variance of the dependent variable is due to the independent variable [32]. Following Cohen [27], $R^2 = .02$ is considered a weak, $R^2 = .13$ a moderate, and $R^2 = .26$ a large effect. Additionally, standard estimation error (*SE*) as well as significance of the entire model (*F*-Test) [33] will be reported.

## Results

### Sample

N = 636 persons (70% female) participated in a cross-sectional online survey. Four hundred fifty-four participants (75% female) between 18 and 73 years old (*M* = 29.91, *SD* = 11.56) were included in the present analyses because they filled out all relevant measures and answered the attention check item correctly. Of those, 272 participants (60%) were in a relationship, 167 participants (37%) reported a mental disorder in the past, and 138 (39%) reported a current mental disorder from whom 110 (80%) were currently in therapeutic treatment. Forty-two participants (9%) reported a suicide attempt in their past. Suicidal ideation in the past two weeks was reported by 98 participants (22%). One hundred twenty-five (28%) participants were students. More specific sociodemographic characteristics can be found in the S1 File.

### Correlation analysis

Regarding the first hypothesis, internal, external entrapment, hopelessness, and suicidal ideation were significantly correlated ($r \geq .50$, $p < .01$; see Table 1). The correlation between suicidal ideation and internal entrapment was significantly stronger than between suicidal

**Table 1. Descriptive statistics and Pearson product-moment correlations.**

|  | *M* | *SD* | *SE* | Range | Med. | Skew. | Kurt. | 1. IE | 2. EE | 3. BHS |
|---|---|---|---|---|---|---|---|---|---|---|
| 1. IE | 8.36 | 8.28 | .39 | 0–24 | 6 | .59 | -1.10 | - |  |  |
| 2. EE | 13.17 | 9.81 | .46 | 0–40 | 11 | .49 | -.81 | .85** | - |  |
| 3. BHS | 6.52 | 5.61 | .26 | 0–20 | 4 | .84 | -.55 | .73** | .68** | - |
| 4. BSS-S | 1.19 | 2.23 | .10 | 0–10 | 0 | 2.05 | 3.58 | .63** | .56** | .72** |

*Note*. IE = Internal Entrapment Subscale; EE = External Entrapment Subscale; BHS = Beck-Hopelessness-Scale; BSS-S = Beck-Suicidal Ideation-Scale-Screen; *M* = mean; *SD* = standard deviation; *SE* = standard error; Med. = Median; Skew. = skewness of the distribution; Kurt. = kurtosis of the distribution

**$p < .01$.

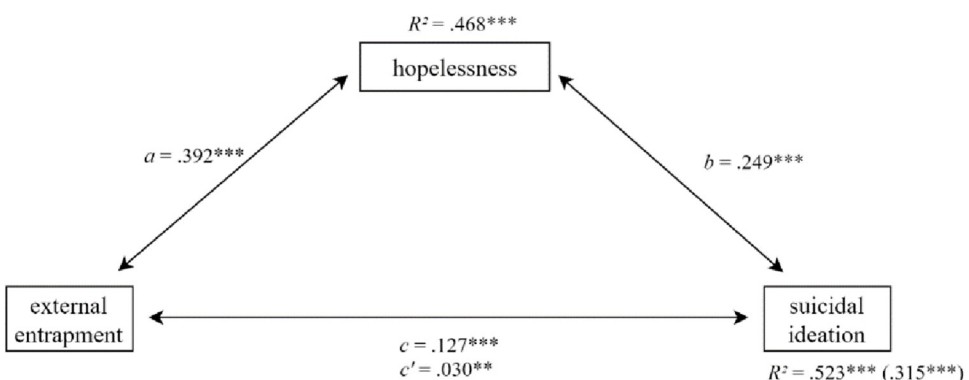

**Fig 2. Mediation model 2 with external entrapment as predictor.**

ideation and external entrapment ($z = 3.422$, $p < .001$). Additionally, hopelessness was significantly stronger correlated with internal than with external entrapment ($z = 2.818$, $p < .002$).

## Mediation analysis

Regarding the second hypothesis, all coefficients were significant ($p < .001$). The effect of internal entrapment on suicidal ideation ($c = .170$, $SE = .013$) was reduced when hopelessness was taken into account as a mediator ($c' = .062$, $SE = .014$) as can be seen in Fig 1. The total effect from external entrapment on suicidal ideation ($c = .127$, $SE = .010$) was also reduced when hopelessness was taken into account as a mediator ($c' = .030$, $SE = .011$; see Fig 2).

There was an indirect effect from internal entrapment to suicidal ideation, which was mediated by hopelessness ($ab = .108$, $SE = .015$, 95%-CI [0.081, 0.138]; see Table 2). The completely standardized indirect effect was $CS_{ab} = .402$. Following Cohen [27], all determination coefficients indicate a great amount of explained variance ($R^2 > .26$) as well as a significant model ($p < .001$). Internal entrapment explained a large proportion of variance in hopelessness (53.4%); correspondingly, the effect was large with $a = .496$ ($SE = .023$). Additionally, explained variance for suicidal ideation increased from 39.8% to 53.9%, when hopelessness was considered as a mediator in this model.

There was also an indirect effect from external entrapment on suicidal ideation, which was mediated by hopelessness ($ab = .097$, $SE = .012$, 95%-CI [0.0757, 0.01217]). The completely

**Table 2. Total, direct, and indirect effects of the mediation models.**

|  | **Effect** | **SE** | **95% CI** | **CS** |
|---|---|---|---|---|
| Mediation model 1 |  |  |  |  |
| Total effect | .170 | .013 | [0.145, 0.195] | .631 |
| Direct effect | .062 | .014 | [0.034, 0.090] | .230 |
| Indirect effect | .108 | .015 | [0.081, 0.138] | .402 |
| Mediation model 2 |  |  |  |  |
| Total effect | .127 | .010 | [0.107, 0.148] | .561 |
| Direct effect | .030 | .011 | [0.008, 0.052] | .132 |
| Indirect effect | .097 | .012 | [0.076, 0.122] | .429 |

*Note.* Mediation model 1: internal entrapment (*X*), hopelessness (*M*), suicidal ideation (*Y*); mediation model 2: external entrapment (*X*), hopelessness (*M*), suicidal ideation (*Y*); *SE* = standard estimation error; 95% CI = 95% confidence interval; *CS* = complete standardized effect.

standardized indirect effect was $CS_{ab} = .429$ and therefore descriptively slightly larger than the effect of internal entrapment on suicidal ideation. The explained variance through external entrapment was slightly lower with 46.8% than in internal entrapment (53.4%). Additionally, explained variance for suicidal ideation increased from 31.5% to 52.3%, when hopelessness was considered as a mediator in the model. This means that the consideration of hopelessness in the mediation model with external entrapment (change in $R^2 = .208$) could explain more variance than in the mediation model with internal entrapment (change in $R^2 = .141$). All determination coefficients including standard estimation errors and F-tests can be found in Table 3.

## Discussion

This study aimed at examining the differential relation between internal and external entrapment and suicidal ideation and investigated whether this relation is (partially) mediated by hopelessness. Prior studies showed that internal entrapment might be more important than external entrapment in the development of suicidal ideation [12, 13]. However, it has not yet been examined why this might be the case. As stated by Rasmussen, Fraser [8], internal entrapment can lead to more hopelessness which in turn can lead to more suicidal ideation than external entrapment. This follows the assumption that internal entrapment might to be more difficult to modify than external entrapment. Due to the immutability, greater feelings of hopelessness might appear. Therefore, hopelessness as a possible mediator for the association between internal entrapment and suicidal ideation could serve as a potential explanation for differential relations between internal and external entrapment and suicidal ideation. We hypothesized that 1) internal and external entrapment, hopelessness and suicidal ideation significantly correlate but that the correlations are larger for internal entrapment than for external entrapment and 2a) both the effects of internal and external entrapment on suicidal

**Table 3. Regression coefficients for the prediction of suicidal ideation.**

| Predictor | | Predicted Variables | | | | | | |
|---|---|---|---|---|---|---|---|---|
| | | *M* (BHS) | | | | *Y* (BSS-S) | | |
| | | Coefficient | SE | *p* | | Coefficient | SE | *p* |
| *X* (IE) | *a* | .496 | .023 | < .001 | *c* | .170 | .013 | < .001 |
| | | $R^2 = .534$ | | | | $R^2 = .398$ | | |
| | | $F(1, 452) = 462.826, p < .001$ | | | | $F(1, 452) = 181.494, p < .001$ | | |
| | | | | | *c'* | .062 | .014 | < .001 |
| *M* (BHS) | | - | - | - | *b* | .218 | .026 | < .001 |
| Constant | *i1* | 2.376 | .202 | < .001 | *i2* | -.744 | .078 | < .001 |
| *X* + *M* | | | | | | $R^2 = .539$ | | |
| | | | | | | $F(2, 451) = 130,088, p < .001$ | | |
| *X* (EE) | *a* | .392 | .019 | < .001 | *c* | .127 | .010 | < .001 |
| | | $R^2 = .468$ | | | | $R^2 = .315$ | | |
| | | $F(1, 452) = 425.563, p < .001$ | | | | $F(1, 452) = 150.183, p < .001$ | | |
| | | | | | *c'* | .030 | .011 | .008 |
| *M* (BHS) | | - | - | - | *b* | .249 | .026 | < .001 |
| Constant | *i1* | 1.363 | .240 | < .001 | *i2* | -.823 | .089 | < .001 |
| *X* + *M* | | | | | | $R^2 = .523$ | | |
| | | | | | | $F(2, 451) = 120.997, p < .001$ | | |

*Note.* IE = internal entrapment subscale; EE = external entrapment subscale; BHS = Beck-Hopelessness-Scale; BSS-S = Beck-Suicidal Ideation-Scale-Screen.

ideation are partially mediated by hopelessness but 2b) this association is greater mediated by hopelessness for internal entrapment.

Results support our first hypothesis, showing strong correlations between internal and external entrapment, hopelessness, and suicidal ideation. As expected, the correlation between internal entrapment and suicidal ideation was stronger than the correlation between external entrapment and suicidal ideation. The same applies to hopelessness, which was stronger correlated with internal than with external entrapment. These findings are in line with Lester [34], who reported descriptively a larger correlation coefficient for internal entrapment and hopelessness as compared to external entrapment and hopelessness, too.

In accordance with hypothesis 2a), both the associations between internal entrapment and suicidal ideation as well as external entrapment and suicidal ideation were partially mediated by hopelessness. In this study, the large amount of explained variance in hopelessness through internal entrapment is striking and emphasizes its role in the development of hopelessness. However, it has to be taken into account that those were only cross-sectional results. These findings should be replicated in a clinical sample with longitudinal data. The role of hopelessness is additionally substantiated by the significant mediation effect suggesting that the effects found for internal entrapment on suicidal ideation [9, 14] can to a certain degree be traced back to its effects on hopelessness. The explained variance in suicidal ideation through internal entrapment is smaller, which emphasizes the role of internal entrapment especially for the development of hopelessness and not to such a large amount of suicidal ideation, which is in line with the high correlations we found for hopelessness and internal entrapment. Earlier studies [12, 13] finding effects for associations between internal entrapment and suicidal ideation did not consider hopelessness, which might explain these effects.

Additionally, hopelessness seems to play an underestimated role for external entrapment as well. Hopelessness mediated the association between external entrapment and suicidal ideation to a greater extent than the association between internal entrapment and suicidal ideation. These findings appear to be contrary to earlier findings [13, 14] that could not demonstrate an effect of external entrapments on suicidal ideation but only for internal entrapment. Also Oakey-Frost, Harris [35] fount that internal but not external entrapment moderated by fearlessness about death could explain the relationship between PTSD and suicidal ideation. Thus, contexts and interindividual differences may be important for the different effects on suicidal ideation. However, the approach differed since we included hopelessness. The findings are still surprising considering arguments that external circumstances might easier be modified than internal entrapment [9, 13]. Still, external entrapment seems to be as much (or maybe even more) associated with hopelessness as internal entrapment. One explanation for this found effect could be the ongoing Covid-19 pandemic. When collecting data from April to July 2021, the pandemic was already ongoing since over a year and vaccinations just started in Germany. Due to several external governmental restrictions in participants' life feelings of external entrapment and hopelessness could have been increased. On the contrary, Bryan, Bryan [36] could not find support for the assumed correlation of physical distancing measures with increased suicide risk in U.S. adults from March until the beginning of April in 2020. However, as the authors stated [36], their findings need to be cautiously interpreted since their cross-sectional study was designed before the Covid-19 pandemic and was conducted at the very beginning of the U.S. pandemic. Therefore, these findings might indicate that the negative effects of physical distancing had not yet occurred but might have been detected later on. Another reason for the results could be that internal and external entrapment are not as different as assumed [11] and/or are not separable enough from hopelessness. Locus of control could also be relevant regarding all three concepts that should be considered.

## Theoretical and practical implications

Given the current results in addition to previous findings, it might be recommendable to consider integrating the differentiation of the subtypes of entrapment into current theories on suicidal ideation, such as the IMV model. Hopelessness, internal as well as external entrapment should be considered and included in clinical risk assessment. The differentiation into internal and external entrapment is not only necessary in regard to the mixed evidence for both constructs but especially because the therapeutic approach for a reduction differs. As mentioned in the study of Oakey-Frost, Harris [35], there might be individual differences responsible for the differential effects of internal and external entrapment but this would make the differentiation even more important to develop interventions for both types. It seems to be important to fulfill individual differences in suicide prevention and risk assessment. Since both external and internal entrapment have been argued as being modifiable [8, 37] the differentiation between internal and external entrapment and, in particular, the need for differently targeted interventions might be important. For internal entrapment the frame of cognitive behavioral therapy seems to be ideal because of the identification and modification of automatic thoughts and beliefs [38, 39]. For external entrapment, behavioral experiments and exposure could be helpful. Those interventions could lead to the feeling of having more control over the external circumstances. This could strengthen participants' expectations of results and hereby maybe reduce feelings of hopelessness. Presumed hopeless situations or states might then not be received as unchangeable but as surmountable. However, these are only theoretical considerations that need to be substantiated by more research in this area. Additionally, two aspects must be noted: First, there is a difference in treating suicidal ideation and preventing the development of suicidal ideation. There are intervention programs specifically targeting suicidality such as the Collaborative Assessment and Management of Suicidality (CAMS) [40] or the Attempted Suicide Short Intervention Program (ASSIP). Second, even though internal and external entrapment might be treatable with different therapeutic techniques, both constructs often occur simultaneously so in clinical practice it might not be easy to treat and target them separately. CAMS represents a treatment that conceptualizes suicidal "drivers", which can include both internal and external entrapment. Those drivers can then be implemented in the individual treatment plan including the knowledge of therapists of particular treatment targeting exactly these drivers [41]. Thus, CAMS targets proximal risk factors could be used in clinical practice.

## Strengths and limitations

A strength of this study was the relatively large community sample with a percentage of participants with a mental disorder comparable to the 12-months prevalence of mental disorders in Germany [42] and a higher prevalence of suicidal ideation than assumed in the general population [43]. Additionally, the use of an attention check item is a strength since it has been shown that this does not only prevent reckless answering behavior but also increases the motivation of participants [44]. Limitations of the study are that there were no control variables, even though there could be other constructs influencing suicidal ideation, the cross-sectional design of the study with its possible biases in the mediation analyses [45], and that this was not a clinical sample and the question about mental disorders was self-report. Additionally, the sample was very young ($M = 29.91$), included 75% females and consisted mainly of well-educated persons (41% A-level/general matriculation standard). Compared to the German population (50.7% females [46], $M = 44.6$ years [47], 33.5% with A-level/general matriculation standard [48]), the generalizability of the study results is limited.

### Future research

Future studies should concentrate on clinical samples. Additionally, fluctuations of suicidal ideation, hopelessness, internal, and external entrapment [49, 50] should be considered. Ecological Momentary Assessment (EMA) offers to repeatedly assess participants in their natural environment [51] capturing moment-to-moment relations between all constructs. This might be a possibility to gain a greater understanding of the complex and dynamic interrelations of risk factors for suicide.

## Conclusion

This study aimed at gaining a deeper understanding for internal and external entrapment and their associations with suicidal ideation. Hopelessness as one potential mediator of these associations was examined. Results of the study emphasize the role of internal entrapment but also highlight hopelessness as a mediator of the association between external entrapment and suicidal ideation. Internal and external entrapment should necessarily be differentiated in suicide risk assessment since the reduction of both (and thereby the prevention of suicidal ideation) might require different interventions. Future studies should examine whether these results can be replicated longitudinally and in a clinical sample.

## Supporting information

**S1 File.**
(DOCX)

**S1 Dataset.**
(XLSX)

## Acknowledgments

We thank Franziska Dienst, Hannah Schiemann, Sarah Schwitzky, Johannes Schroers and Jessica Struchhold for support in collecting data.

We acknowledge support by the Open Access Publication Fund of the University of Duisburg-Essen.

## Author Contributions

**Conceptualization:** Inken Höller, Dajana Schreiber, Thomas Forkmann.

**Data curation:** Amelie Kremers.

**Formal analysis:** Inken Höller, Amelie Kremers, Dajana Schreiber.

**Investigation:** Inken Höller, Amelie Kremers.

**Methodology:** Inken Höller, Dajana Schreiber, Thomas Forkmann.

**Project administration:** Inken Höller, Dajana Schreiber, Thomas Forkmann.

**Resources:** Dajana Schreiber, Thomas Forkmann.

**Supervision:** Thomas Forkmann.

**Writing – original draft:** Inken Höller.

**Writing – review & editing:** Amelie Kremers, Dajana Schreiber, Thomas Forkmann.

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
