## [Decision Letter · Decision Letter 0]

18 Mar 2022

PONE-D-22-04773Trapped in my Inner Prison – Cross-Sectional Examination of Internal and External Entrapment, Hopelessness and Suicidal IdeationPLOS ONE

Dear Dr. Höller,

Thank you for submitting your manuscript to PLOS ONE. After careful consideration, we feel that it has merit but does not fully meet PLOS ONE’s publication criteria as it currently stands. Therefore, we invite you to submit a revised version of the manuscript that addresses the points raised during the review process.

There several issues pertaining both adequate analysis and interpretation of results that both reviewers have highlighted e need attention.

We look forward to receiving your revised manuscript.

Kind regards,

Pedro Vieira da Silva Magalhaes, M.D., Ph.D.

Academic Editor

PLOS ONE

Journal Requirements:

 [This study did not receive funding.]

Reviewers' comments:

Reviewer's Responses to Questions

**Comments to the Author**

1. Is the manuscript technically sound, and do the data support the conclusions?

Reviewer #1: Partly

Reviewer #2: Yes

2. Has the statistical analysis been performed appropriately and rigorously? 

Reviewer #1: Yes

Reviewer #2: Yes

3. Have the authors made all data underlying the findings in their manuscript fully available?

Reviewer #1: Yes

Reviewer #2: Yes

4. Is the manuscript presented in an intelligible fashion and written in standard English?

Reviewer #1: Yes

Reviewer #2: Yes

5. Review Comments to the Author

Reviewer #1: Thank you for the opportunity to review your manuscript! I am fascinated by the ongoing IMV debate, specifically regarding the specific constructs hypothesized to play a role in suicide phenomenology. This study makes and important contribution thereto and has interesting implications moving forward. However, while this study is standard in its methodological approach, it is also significantly limited by a cross-sectional mediation design and may benefit from the inclusion of an additional mode. Finally, there are several bold but unsubstantiated claims in the manuscript that will require further contextualization, discussion, and most importantly, citation. Please see attached document for full review and comments.

Reviewer #2: Review for manuscript PONE-D-22-04773: Trapped in my Inner Prison – Cross-Sectional Examination of Internal and External Entrapment, Hopelessness and Suicidal Ideation

This is an interesting study. It is limited by a non-clinical cross-sectional sample, although the authors point out these limitations.

Methods

The Sample results (p.5, l. 105-112) should be reported in the Results section.

More information about the sample would be useful, such as education level, socio-economic status, etc. What proportion were students recruited from the University?

Measures

How were past and current mental disorders measured?

The scoring of the BSS is not clearly explained (sentence on p.6, l.136-137). How were the five items which were used scored?

Results

In Table 1 it seems that the Min and Max are the possible min. and max. values for the scale, rather than the results obtained from the sample. These should be the min. and max. values from the sample. If the distributions were very skewed, median and interquartile range should be reported too.

The means obtained for all the psychological measures (Table 1) are fairly low, as this is not a clinical sample. Could the authors please comment on how skewed the distributions were and whether this could have implications for the statistical methods used. Often with non-clinical samples the distributions of psychological scales are extremely skewed.

Discussion

The sentence on p. 12, l. 250-251 makes an inference about the role of internal entrapment on the development of hopelessness, however this is a cross sectional non-directional study.

Although hopelessness mediated the association between external entrapment and suicidal ideation to a slightly greater extent, the results are very similar to those of internal attachment. Because the results are so similar, and without testing for significant differences of the explained variance (R2) and the coefficients, it is difficult to draw a definitive conclusion that one mediation is stronger than the other.

Strengths and Limitations

The authors mention the use of an “attention check item” (p.14, l.294), but this is not included in the Measures section.

6. PLOS authors have the option to publish the peer review history of their article (what does this mean?). If published, this will include your full peer review and any attached files.

Reviewer #1: **Yes: **D Nicolas Oakey-Frost

Reviewer #2: No

---

## [Author Response · Author response to Decision Letter 0]

20 Apr 2022

Reviewer #1

This is an interesting study. It is limited by a non-clinical cross-sectional sample, although the authors point out these limitations.

Methods

The Sample results (p.5, l. 105-112) should be reported in the Results section. 

Response: We consider this sample information necessary to provide insight into the composition of the sample at this point. The sample information is not reported in the results section because the results section should mainly highlight the results of the statistical analyses that were conducted to test our hypotheses (i.e., correlation and mediation analysis). We hope that it is okay if we leave it like that.

More information about the sample would be useful, such as education level, socio-economic status, etc. Response: sociodemographic characteristics are summed up in a table (Table 1 in supplementary material), which can be found in the supplementary material. Information on this table have been added to the participants’ section.

What proportion were students recruited from the University? 

Response: The proportion of students from the University of Duisburg-Essen was not measured per se but the proportion of (university) students in general (i.e., irrespective of at which university they were enrolled) was 28% (n= 125). We added this information to the participants’ section.

Measures

How were past and current mental disorders measured? 

Response: Participants were asked with single items each whether they had a mental disorder in the past or at the moment and whether they received treatment. We are aware that this is only self-report, this is why we used the wording “participants reported a mental disorder”. We added the self-report nature to the limitations.

The scoring of the BSS is not clearly explained (sentence on p.6, l.136-137). How were the five items which were used scored? 

Response: The (total) BSS score ranges from 0 to 38 and is formed from the sum of the first 19 items. The last two items (20 and 21) describe former suicide attempts (and not thoughts) and are usually not included in the total score. However, because of the ambiguous factor structure of the total BSS score, the BSS-Screen was used to describe suicidal ideation. In contrast to the total BSS-Score, the BSS-Screen score has a unidimensional factor structure. This BSS-Screen includes the sum score of only the first five items of the BSS. We added information on this to the manuscript.

Results

In Table 1 it seems that the Min and Max are the possible min. and max. values for the scale, rather than the results obtained from the sample. These should be the min. and max. values from the sample. If the distributions were very skewed, median and interquartile range should be reported too. 

Response: Thank you for this suggestion. The “Min” and “Max” were replaced by the range, so it is easier to understand that those are the values from the sample. Information on skewness, kurtosis, range and median have also been added to the table. We hope this meets your expectations.

The means obtained for all the psychological measures (Table 1) are fairly low, as this is not a clinical sample. Could the authors please comment on how skewed the distributions were and whether this could have implications for the statistical methods used. Often with non-clinical samples the distributions of psychological scales are extremely skewed. 

Response: We followed your suggestion above and added skewness and kurtosis to the table. Due to the large sample and the fact that bootstrapping does not depend on normal distribution, the skewness and kurtosis has not been considered to impact validity of the results any further. 

Discussion

The sentence on p. 12, l. 250-251 makes an inference about the role of internal entrapment on the development of hopelessness, however this is a cross sectional non-directional study. 

Response: You are completely right, that this is a limitation and. We changed the wording and added the limitations of the study, so it now reads like this: “In this study, the large amount of explained variance in hopelessness through internal entrapment is striking and emphasizes its role in the development of hopelessness. However, it has to be considered that those were only cross-sectional results. These findings should be replicated in a clinical sample with longitudinal data.” We hope that this meets your expectations.

Although hopelessness mediated the association between external entrapment and suicidal ideation to a slightly greater extent, the results are very similar to those of internal attachment. Because the results are so similar, and without testing for significant differences of the explained variance (R2) and the coefficients, it is difficult to draw a definitive conclusion that one mediation is stronger than the other.

Response: We agree. However, Internal and external entrapment being so similar in its effect on hopelessness / SI is still surprising considering previous findings in which internal entrapment seems to have a more important role with regard to suicidality (Höller et al., 2021; O’Connor & Portzky, 2018; Owen et al., 2018; Rasmussen et al., 2010) . Indeed, it is a limitation that there was no testing for significance of the explained variances, coefficients and standardized effects. Anyway, the completely standardized effect is a proper measure to compare effects from different mediation models (Hayes, 2017), the size of the effect itself is already a surprising and relevant result. 

Strengths and Limitations

The authors mention the use of an “attention check item” (p.14, l.294), but this is not included in the Measures section. 

Response: The attention check item is already mentioned in the Methods (sample) section. We decided not to repeat this information in the measures section, as this item only served to filter out those people who completed the survey responsibly. Therefore, the attention check item is not a measurement per se. Participants who did not correctly answer the attention check item were excluded from the data analyses. We added the following sentences to the manuscript: “The attention check item asked participants to check the answer option “4”. In case they did not check this item correctly due to inattentiveness, they were excluded from analyses.” In the participants section is statet that 454 participants answered the attentiom check item correctly and completed all relevant questionnaires. We hope this answers your question.

Reviewer #2

Thank you for the opportunity to review your manuscript! I am fascinated by the ongoing IMV debate, specifically regarding the specific constructs hypothesized to play a role in suicide phenomenology. This study makes and important contribution thereto and has interesting implications moving forward. However, while this study is standard in its methodological approach, it is also significantly limited by a cross-sectional mediation design and may benefit from the inclusion of an additional mode. Finally, there are several bold but unsubstantiated claims in the manuscript that will require further contextualization, discussion, and most importantly, citation:

General comments:

I do understand and appreciate that the authors may consider English as a second language. However, there were points where the narrative was unclear and could benefit from copyediting. I was able to understand the thrust, but re-reading and re-evaluating readability will help make the manuscript clearer for the reader.

Response: The manuscript was additionally proofread, we hope that this now meets your expectations.

The authors should refrain from using the term “confirmed” when referencing their own hypotheses or the results of extant literature. The nature of parametric statistics precludes the confirmation of results and only support for the observed results. 

Response: The corresponding phrase (line 249) was changed to “Results support for…”. We additionally searched the manuscript for the term “confirm” and changed it when used inappropriately. Now the word confirmed is only used for other studies. We did not mean to confuse the reader or to give wrong information. 

Background

Generally, when introducing a cited study please briefly describe the sample and the design of the cited study. This will give the reader more context for the results being described 

Response: Short information about the design and sample of the cited studies was added.

Lines 73 – 74 will require a citation for the assertion that “research...has revealed that internal entrapment seems to be more important...” 

Response: We are sorry, there was a reference missing. This information came from O'Connor & Portzky (2018). We added this citation/reference.

Please briefly describe the findings of Owen, Dempsey et al., more in depth 

Response: Information about the responding design and sample of this study was added, as well as its outcome distinguishing between internal and external entrapment. 

Lines 78 – 79 appears to be an assertion that comes out of the blue; however, it becomes clearer when discussing this assertion in the Discussion section of the manuscript, lines 230 – 233. Clarifying this assertion in the Introduction would be useful 

Response: The explanations made in the discussion have been added to the Introduction. We hope everything is now easier to understand.

Line 97; although I understand the reflex to argue from a perspective of reason, it is not sufficient to argue for the differential effects of entrapment dimensions on hopelessness based on reason; reasonable based on...? The Cry of Pain model? Johnson, Gooding? These at least should be cited and expanded upon a bit more 

Response: The word “reasonable” was changed to “conceivable” to support the theoretical assertion which was made according to the empirical findings mentioned above. We also added that this assumption is based on the above mentioned studies and considerations.

Measures

Please provide a detailed breakdown of demographics in a table to further contextualize the sample; among the variables already include, likely additions should include racial identity, gender identity, and the baseline risk variables (suicide attempt, suicidal ideation, etc.)

Response: We already give information on suicide attempt in the past as well as suicidal ideation and gender identity. However, we also added a table (Table 1 in supplementary material) including some more sociodemographic characteristics and risk variables as supplementary material. We hope that this meets your expectations. However, there is no information on racial identity; it is not common to assess this in Germany.

Please provide further detail as to whether measures were sum scored or mean scored 

Response: all measures were sum scored, which has been added to the descriptions. 

Please provide the following estimates in the correlation table in addition to those provided: standard error (SE), skew, kurtosis, range, for each of the constructs measured 

Response: The missing information has been added to Table 1 in the manuscript.

Statistical analyses

Line 165; the assertion of either full or partial mediation is contrary to the hypotheses and strikes of HARKing; please clarify here and in the hypotheses whether you predict full or partial mediation, not both. 

Response: The part of the sentence in which the full mediation was mentioned was deleted to clarify our partial mediation hypothesis. We apologize for this inconvenience. We do not know how this happened; it might have happened during the translation process but we clearly did not conduct HARKing, which is quite a strong accusation. Our hypotheses were formulated before data collection even began.

It seems reasonable that the authors should test the hypothesized model with the Entrapment Scale total score considering their intention to determine whether internal vs. external entrapment explain more variance in SI as explained by hopelessness; this would make for a potentially stronger paper by match some of their cited literature (i.e., Littlewood, Gooding), testing an inference of the IMV, and align with their assertion that “it remains unclear how and why internal and external entrapment influence suicidal ideation”. Testing all three models would permit more nuanced discussion of theoretical implications. 

Response: Our intention above all was to investigate the difference between internal and external entrapment relating to suicidal ideation and hopelessness. The missing differentiation of internal and external entrapment in the IMV model was one of our motivations for this research. Along with our hypotheses, we expected the effect of internal entrapment to be greater. To examine our hypotheses it was necessary to conduct the models separately for internal and external entrapment. Due to your suggestions, we conducted post-hoc a third mediation model including total entrapment as a predictor. We added these post-hoc analyses to the statistical analyses section and provide the results in the supplementary material. All coefficients were significant (p < .001. Overall, when comparing the results of total entrapment to those of the preceding analyses, the completely standardized indirect effect of total entrapment was smaller than the completely standardized indirect effect of external entrapment (for more detailed results see supplementary material. This suggests that the role of external entrapment needs to be further examined independently of internal entrapment especially because despite the statistics the treatment differs (s. comment below).

Discussion

Line 263; please compare and contrast how your findings are contrary to those of the cited studies 

Response: Our findings are contrary to those of Höller et al. (2021) and Lucht et al. (2020), as our results show an effect of external entrapment on suicidal ideation. We added discussion on this topic to the manuscript.

Lines 267 – 274; these are major assumptions regarding the impact of the COVID-19 pandemic on suicidal ideation and external entrapment that will require contextualization within the available literature. For example, Bryan, Bryan, and Baker (2021) found no relationship between physical distancing and with worse mental health outcomes. This observation has implications for the relationship between the COVID-19 pandemic and external entrapment. 

Response: Thank you for this suggestion. Unfortunately, we cannot find the paper you mentioned and are therefore not exactly sure what you mean. If you could specify, we would be happy to adapt the manuscript accordingly. Physical distancing is not the same as feeling externally entrapped because of the governments’ social distancing rules. Feeling entrapped in this case does not necessarily mean to actually be trapped in the sense of physical distance but more the feeling of being trapped due to the rules and no hope that this will change any time in the near future. But we could further adapt the manuscript to the paper on physical distancing when we know which paper exactly you mean. We made some changes to weaken our conclusions and hope this meets your expectations now.

Line 278; given that both external and internal entrapment in relation to suicidal ideation were significantly explained by hopelessness to some extent, I am not convinced that these constructs require separation. 

Response: Both internal and external entrapment were mentioned here to emphasize the connection with suicidal ideation and hopelessness. We added one more sentences on the differentiation of internal and external entrapment. We are not sure whether we understand your point correctly. Statistically, internal and external entrapment are highly correlated but yet distinct constructs. In this study, the correlation between internal and external entrapment is high with .85. However, the entrapment scale was designed to assess both internal and external entrapment and study results that examined differences in internal and external entrapment found different associations of the constructs with suicidal ideation. There is a difference whether your patient has rumination (internal entrapment) or whether he/she is hopeless due to external circumstances (job loss, relationship problems � external entrapment). The two constructs measure different states that need different treatment approaches. Those treatment approaches have been discussed in the discussion section, too. The separation of these constructs is only a consequence of the mixed findings. Even if both were equally important in the development of suicidal ideation, the interventions would differ. For clinical practice and for suicide prevention, this needs to be considered. Both construcs are important in the development of suicidal ideation but it would be interesting to investigate how different treatment approaches could prevent individuals from developing suicidal ideation.

Their conclusions may be supported by inclusion of the proposed total score model. If a worse model fit is observed in contrast to the internal or external entrapment models, this conclusion would be significantly strengthened 

Response: We foresee from a third model since this was not our goal. Our goal was to examine differential effects from internal and external entrapment on suicidal ideation. Especially because this differentiation is especially important when talking about possible interventions see our comment above. However, we did follow your suggestion and added a third model to the manuscript. This is mentioned as post-hoc analysis in the statistical anylses section. Results can be found in the supplementary material.

Additionally, the authors may wish to discuss the findings in different contexts; for example, they may wish to reference the findings of Oakey-Frost et al., 2021 who found that internal but not external entrapment with fearlessness about death explained the relationship between PTSD and suicidal ideation. Thus, the effect of one vs. the other may vary according to important interindividual differences 

Response: These important findings were added to the manuscript. Thank you for this suggestion. However, we would like to mention, that, in our view, this underlines again the importance of differentiating between these constructs. Even if both constructs were equally important, the importance might differ depending on interindividual differences. Maybe there is no “one fits all”-model.

: 

Lines 279 – 278; the argument for differential treatment approaches for internal and external entrapment is potentially reasonable, but unsubstantiated with regard to approaches to external entrapment specifically. 

Response: In fact, there are no evidence-based interventions yet. That is why we provide some theoretical considerations here. We added a sentence that those are only theoretical considerations and that more empirical research in this area is needed.

Additionally, the question must be asked: are we aiming to improve entrapment or suicidal thoughts and behaviors? Other viable treatment options do exist and should be discussed; The Collaborative Assessment and Management of Suicidality (CAMS; Jobes, 2017) is a framework used to target entrapment OR internal vs. external entrapment as the driver of suicidal thoughts and behaviors based on the needs of the individual in question. This critique includes lines 311 – 312. Thus, the authors may be correct that internal vs. external entrapment may require different approaches based on the needs of the individual and their suicidal driver. But this claim has to be substantiated with citation and discussion. 

Response: We are aware that there are intervention programs for patients regarding suicidality but as you said there is a difference of treating suicidality and focusing on proximal risk factors. Intervention targeting proximal risk factors would aim at preventing patients from even developing suicidal ideation. We added reference to the CAMS to the manuscript. We added discussion on this to the manuscript. We also added to the manuscript that we have to differentiate between programs that treat suicidality and interventions that target proximal risk factors. We hope that this meets your expectations now. 

Finally, the authors should discuss the limitations of cross-sectional mediation in the context of Maxwell & Cole, 2007. 

Response: Maxwell, Cole and Mitchell (2011) was added to the limitations as this article discusses partial mediation models in cross-sectional studies. Thank you for this suggestion.

Figure

I would prefer that arrows in the models are bidirectional. While the conclusions may be theoretically supported, inferring directionality and therefore temporal relationships is not supported in cross-sectional mediation models. 

Response: We amended the mediation models but hope that this is not misleading.

Thank you for allowing me to review your manuscript and for your valuable contribution to the suicide prevention literature base. I hope that my comments and feedback are of value to the authors and to the editor(s) of the journal. I genuinely appreciate the opportunity to engage in this valuable scientific discourse.

Höller, I., Rath, D., Teismann, T., Glaesmer, H., Lucht, L., Paashaus, L., Schönfelder, A., Juckel, G., & Forkmann, T. (2021). Defeat, entrapment, and suicidal ideation: Twelve‐month trajectories. Suicide and Life‐Threatening Behavior. 

O’Connor, R. C., & Portzky, G. (2018). The relationship between entrapment and suicidal behavior through the lens of the integrated motivational–volitional model of suicidal behavior. Curr Opin Psychol, 22, 12-17. https://doi.org/10.1016/j.copsyc.2017.07.021

Owen, R., Dempsey, R., Jones, S., & Gooding, P. (2018). Defeat and entrapment in bipolar disorder: exploring the relationship with suicidal ideation from a psychological theoretical perspective. Suicide and Life‐Threatening Behavior, 48(1), 116-128. https://doi.org/10.1111/sltb.12343

Rasmussen, S. A., Fraser, L., Gotz, M., MacHale, S., Mackie, R., Masterton, G., McConachie, S., & O'Connor, R. C. (2010, 3/2010). Elaborating the cry of pain model of suicidality: Testing a psychological model in a sample of first-time and repeat self-harm patients. British Journal of Clinical Psychology, 49(1), 15-30. https://doi.org/10.1348/014466509X415735

---

## [Decision Letter · Decision Letter 1]

13 May 2022

PONE-D-22-04773R1Trapped in my Inner Prison – Cross-Sectional Examination of Internal and External Entrapment, Hopelessness and Suicidal IdeationPLOS ONE

Dear Dr. Höller,

Thank you for submitting your manuscript to PLOS ONE. After careful consideration, we feel that it has merit but does not fully meet PLOS ONE’s publication criteria as it currently stands. Therefore, we invite you to submit a revised version of the manuscript that addresses the points raised during the review process.

Both reviewers have indicated the manuscript is greatly improved with some minor editing and revisions remaining.

We look forward to receiving your revised manuscript.

Kind regards,

Pedro Vieira da Silva Magalhaes, M.D., Ph.D.

Academic Editor

PLOS ONE

Journal Requirements:

Reviewers' comments:

Reviewer's Responses to Questions

**Comments to the Author**

1. If the authors have adequately addressed your comments raised in a previous round of review and you feel that this manuscript is now acceptable for publication, you may indicate that here to bypass the “Comments to the Author” section, enter your conflict of interest statement in the “Confidential to Editor” section, and submit your "Accept" recommendation.

Reviewer #1: (No Response)

Reviewer #2: (No Response)

2. Is the manuscript technically sound, and do the data support the conclusions?

Reviewer #1: Yes

Reviewer #2: Yes

3. Has the statistical analysis been performed appropriately and rigorously? 

Reviewer #1: Yes

Reviewer #2: Yes

4. Have the authors made all data underlying the findings in their manuscript fully available?

Reviewer #1: Yes

Reviewer #2: Yes

5. Is the manuscript presented in an intelligible fashion and written in standard English?

Reviewer #1: Yes

Reviewer #2: Yes

6. Review Comments to the Author

Reviewer #1: Many thanks to the authors for their thoughtful responses to the feedback. I have just a few more comments that I think would be best addressed prior to publication:

Background

Line 61, 64, and 67 use the word “confirm[ed]” in relation to the results of the cited studies. I don’t want to sound like a broken record, but “confirming” entrapment as a risk factor in a cross-sectional mediation analysis is a mischaracterization. I think it is incredibly important to not overstate the implications of any results, especially to the general public.

Discussion

In previous feedback, the authors indicated their willingness to discuss differential findings with relation to social distancing and external entrapment in the context of the COVID-19 pandemic. Unfortunately, I provided the wrong year for the citation I included. Would the authors be willing to Discuss their findings from Lines 287 – 297 in the context of the following citation:

Bryan, Bryan, & Baker. (2020). Associations among state-level physical distancing measures and suicidal thoughts and behaviors among U.S. adults during the early COVID-19 pandemic. Suicide and Life-Threatening Behavior.

To first put the following comment in context, I apologize if my previous feedback was not clearly articulated. I want to be clearer about where I am coming from:

Line 310, their assertion that “there have been no evidence-based interventions for the reduction of entrapment” is where I would recommend, they discuss CAMS. CAMS targets suicidal ideation and suicidal behavior via conceptualization of the patient’s “drivers” which indeed may include perceptions of internal entrapment and/or external entrapment. The clinician then uses their knowledge to recommend particular treatment orientations to target the driver. Thus, treating suicidal ideation occurs via treatment of the factors that lead to suicidal ideation. Additionally, they argue that CAMS and ASSIP do not target proximal risk; from a CAMS perspective, it does. The first session of CAMS includes collaborative formulation of a Safety/Crisis Prevention Plan; these interventions are inherently designed to target “proximal risk” (Line 323).

In essence, I agree with the authors: internal vs. external entrapment may be differentiated based on their treatment needs. I also believe the authors have a good opportunity to make their argument in the context of available interventions. Treatment may be accomplished via CAMS which, again, identifies a driver (e.g., internal entrapment) and then uses patient specific interventions to target the driver (e.g., cognitive therapy [see Jobes, 2017]).

Overall, I genuinely appreciate the author’s responses and persistence thus far. Thank you again for the opportunity to review your manuscript.

Reviewer #2: The authors have responded to comments and improved the paper. Just some minor comments remain.

Methods

The Sample results (p.5, l. 105-112) should be reported in the Results section.

Response: We consider this sample information necessary to provide insight into the composition of the sample at this point. The sample information is not reported in the results section because the results section should mainly highlight the results of the statistical analyses that were conducted to test our hypotheses (i.e., correlation and mediation analysis). We hope that it is okay if we leave it like that.

Reviewer response: The standard is to report the details of the collected sample in the Results, rather than the Methods.

Measures

How were past and current mental disorders measured?

Response: Participants were asked with single items each whether they had a mental disorder in the past or at the moment and whether they received treatment. We are aware that this is only self-report, this is why we used the wording “participants reported a mental disorder”. We added the self-report nature to the limitations.

Reviewer response: The details about this question should be included in the Measures section.

The scoring of the BSS is not clearly explained (sentence on p.6, l.136-137). How were the five items

which were used scored?

Response: The (total) BSS score ranges from 0 to 38 and is formed from the sum of the first 19 items. The last two items (20 and 21) describe former suicide attempts (and not thoughts) and are usually not included in the total score. However, because of the ambiguous factor structure of the total BSS score, the BSS-Screen was used to describe suicidal ideation. In contrast to the total BSS-Score, the BSS-Screen score has a unidimensional factor structure. This BSS-Screen includes the sum score of only the first five items of the BSS. We added information on this to the manuscript.

Reviewer response: The sentence on p.7, l. 148-149 of the revised manuscript is still unclear English and needs editing: “The first 19 items are summed up in case items 4 and 5 have been answered at least 1.”

In addition to the above points, a further limitation of the study is that respondents tended to be predominantly young, and perhaps with a large proportion of university and tertiary educated persons, although this will depend on what the standard for Germany is. A brief comment on how the sample compares to the German population may be useful.

Another comment I have is that internal and external entrapment are so highly related that they would likely often occur together and may not be always easy to separate or be treated separately in psychological interventions.

7. PLOS authors have the option to publish the peer review history of their article (what does this mean?). If published, this will include your full peer review and any attached files.

Reviewer #1: No

Reviewer #2: No

---

## [Author Response · Author response to Decision Letter 1]

14 Jun 2022

Reviewer #1

Many thanks to the authors for their thoughtful responses to the feedback. I have just a few more comments that I think would be best addressed prior to publication:

Background

Line 61, 64, and 67 use the word “confirm[ed]” in relation to the results of the cited studies. I don’t want to sound like a broken record, but “confirming” entrapment as a risk factor in a cross-sectional mediation analysis is a mischaracterization. I think it is incredibly important to not overstate the implications of any results, especially to the general public. 

Response: We are sorry that you got the impression we wanted to overstate the findings. To not overrate the results of the cited study the term “confirmed” was changed into “shown” (line 61), “pointed to” (line 64) and “demonstrated” (line 67/70). 

Discussion

In previous feedback, the authors indicated their willingness to discuss differential findings with relation to social distancing and external entrapment in the context of the COVID-19 pandemic. Unfortunately, I provided the wrong year for the citation I included. Would the authors be willing to Discuss their findings from Lines 287 – 297 in the context of the following citation: 

Bryan, Bryan, & Baker. (2020). Associations among state-level physical distancing measures and suicidal thoughts and behaviors among U.S. adults during the early COVID-19 pandemic. Suicide and Life-Threatening Behavior.

Response: This reference was added to the manuscript (lines 294-300). We discussed the findings of Bryan et al. (2020) that there were no correlations between physical distancing and suicidal ideation, however, as the authors state themselves, they collected data at the very beginning of the pandemic, while we collected data when the pandemic was already going on for more than a year.

To first put the following comment in context, I apologize if my previous feedback was not clearly articulated. I want to be clearer about where I am coming from:

Line 310, their assertion that “there have been no evidence-based interventions for the reduction of entrapment” is where I would recommend, they discuss CAMS. CAMS targets suicidal ideation and suicidal behavior via conceptualization of the patient’s “drivers” which indeed may include perceptions of internal entrapment and/or external entrapment. The clinician then uses their knowledge to recommend particular treatment orientations to target the driver. Thus, treating suicidal ideation occurs via treatment of the factors that lead to suicidal ideation. Additionally, they argue that CAMS and ASSIP do not target proximal risk; from a CAMS perspective, it does. The first session of CAMS includes collaborative formulation of a Safety/Crisis Prevention Plan; these interventions are inherently designed to target “proximal risk” (Line 323). 

Response: CAMS is now named to be an effective treatment that is addressing internal and/or external entrapment. We extended the paragraph on CAMS starting on line 334-340. We hope that this meets your expectations.

In essence, I agree with the authors: internal vs. external entrapment may be differentiated based on their treatment needs. I also believe the authors have a good opportunity to make their argument in the context of available interventions. Treatment may be accomplished via CAMS which, again, identifies a driver (e.g., internal entrapment) and then uses patient specific interventions to target the driver (e.g., cognitive therapy [see Jobes, 2017]). 

Response: We agree with you that is why we added CAMS as a possible treatment (See comment above)

Overall, I genuinely appreciate the author’s responses and persistence thus far. Thank you again for the opportunity to review your manuscript.

Reviewer #2

The authors have responded to comments and improved the paper. Just some minor comments remain.

Methods

The Sample results (p.5, l. 105-112) should be reported in the Results section.

Response: We consider this sample information necessary to provide insight into the composition of the sample at this point. The sample information is not reported in the results section because the results section should mainly highlight the results of the statistical analyses that were conducted to test our hypotheses (i.e., correlation and mediation analysis). We hope that it is okay if we leave it like that.

Reviewer response: The standard is to report the details of the collected sample in the Results, rather than the Methods. 

Response: We think that this actually depends on the journal but we understand where you are coming from so we followed your suggestion. The information of the sample results is now reported in the Results section. The sub-headlines “Correlation Analysis” and “Mediation Analysis” have been added to the Results section to obtain a better structure. 

Measures

How were past and current mental disorders measured?

Response: Participants were asked with single items each whether they had a mental disorder in the past or at the moment and whether they received treatment. We are aware that this is only self-report, this is why we used the wording “participants reported a mental disorder”. We added the self-report nature to the limitations.

Reviewer response: The details about this question should be included in the Measures section.

Response: The information delivered above was added to the Measures section. 

The scoring of the BSS is not clearly explained (sentence on p.6, l.136-137). How were the five items

which were used scored?

Response: The (total) BSS score ranges from 0 to 38 and is formed from the sum of the first 19 items. The last two items (20 and 21) describe former suicide attempts (and not thoughts) and are usually not included in the total score. However, because of the ambiguous factor structure of the total BSS score, the BSS-Screen was used to describe suicidal ideation. In contrast to the total BSS-Score, the BSS-Screen score has a unidimensional factor structure. This BSS-Screen includes the sum score of only the first five items of the BSS. We added information on this to the manuscript.

Reviewer response: The sentence on p.7, l. 148-149 of the revised manuscript is still unclear English and needs editing: “The first 19 items are summed up in case items 4 and 5 have been answered at least 1.”

Response: We are sorry the reformulation did not meet your expectation. We tried again and also asked an independent person whether it is understandable how the sum score was build. The sentences now read like this in the manuscript: When participants answered items 4 and 5 (screening items) with at least 1, the first 19 items were used to build a sum score. Otherwise, participants had a sum score of zero. 

In addition to the above points, a further limitation of the study is that respondents tended to be predominantly young, and perhaps with a large proportion of university and tertiary educated persons, although this will depend on what the standard for Germany is. A brief comment on how the sample compares to the German population may be useful. 

Response: Information of the German population was added to the Strengths and Limitations – Section (lines 347-351). We emphasized that our sample includes a higher female percentage, is younger, and a higher percentage of high education than the general German population. We also stated in the limitation section that this limits the generalizability of our findings.

Another comment I have is that internal and external entrapment are so highly related that they would likely often occur together and may not be always easy to separate or be treated separately in psychological interventions. 

Response: Indeed, internal entrapment and external entrapment are highly related and often occur together. We modified the discussion and emphasize that it might not be easy to separate them in clinical practice and added more to the treatment program CAMS that targets both internal and external entrapment.

---

## [Editor Report · Decision Letter 2]

22 Jun 2022

Trapped in my Inner Prison – Cross-Sectional Examination of Internal and External Entrapment, Hopelessness and Suicidal Ideation

PONE-D-22-04773R2

Dear Dr. Höller,

We’re pleased to inform you that your manuscript has been judged scientifically suitable for publication and will be formally accepted for publication once it meets all outstanding technical requirements.

Kind regards,

Pedro Vieira da Silva Magalhaes, M.D., Ph.D.

Academic Editor

PLOS ONE
---

## [Editor Report · Acceptance letter]

5 Jul 2022

PONE-D-22-04773R2 

Trapped in my Inner Prison – Cross-Sectional Examination of Internal and External Entrapment, Hopelessness and Suicidal Ideation 

Dear Dr. Höller:

I'm pleased to inform you that your manuscript has been deemed suitable for publication in PLOS ONE. Congratulations! Your manuscript is now with our production department. 

Kind regards, 

on behalf of

Professor Pedro Vieira da Silva Magalhaes 

Academic Editor

PLOS ONE